# Effect of a High-Temperature Treatment on Structural-Phase State and Mechanical Properties of IMC of the Ti-25Al-25Nb at.% System

**DOI:** 10.3390/ma15165560

**Published:** 2022-08-12

**Authors:** Mazhyn Skakov, Yernat Kozhakhmetov, Nurya Mukhamedova, Arman Miniyazov, Igor Sokolov, Azamat Urkunbay, Gainiya Zhanbolatova, Timur Tulenbergenov

**Affiliations:** 1National Nuclear Center of the Republic of Kazakhstan, Kurchatov KZ-071100, Kazakhstan; 2Institute of Atomic Energy Branch of the National Nuclear Center of the Republic of Kazakhstan, Kurchatov KZ-071100, Kazakhstan

**Keywords:** titanium aluminides, Ti-Al-Nb, microstructure, mechanical properties, high-temperature treatment, spark-plasma sintering, intermetallic compound

## Abstract

In this research, samples of an alloy with a bimodal structure were studied on the basis of a previously developed technology for obtaining hydrogen storage materials based on the Ti-Al-Nb system. The results of SPS of mechanically activated powder mixtures of the Ti-Al-Nb system at a temperature of 1300 °C make it possible to obtain an alloy with a predominantly bimodal structure. However, an insignificant presence of TiAl_3_, AlNb_2_ phases, and unreacted niobium is still observed in the structure. The mechanical properties of alloys of the Ti-Al-Nb system after sintering show abnormally low values of strength and ductility (less than 150 MPa). Two-stage heat treatment of alloys of the Ti-Al-Nb system leads to the decomposition of large precipitates of TiAl_3_ with the formation of O-phase nuclei, as well as to the complete dissolution of unreacted niobium and AlNb_2_ phases. Heat treatment of alloys of the Ti-Al-Nb system contributes to an increase in its strength by approximately 10 times (1310 MPa, MA-180), and ductility by 2 times (1322 MPa, MA-20). The surface fracture of samples obtained after testing is characterized by intergranular (intercrystalline) brittle fracture with “river” or “step” features.

## 1. Introduction

In the last decade, interest in intermetallic compounds and alloys based on them has been continuously growing, which is due to the huge potential of a number of these materials in the manufacture of structures operating under severe external loading conditions, including high-temperature treatment [1,2,3,4]. Of particular interest are titanium aluminides due to their low density and, consequently, high specific strength [5,6,7]. However, in the normal temperature range, titanium aluminides, like most intermetallic compounds, are characterized by increased brittleness in the polycrystalline state, which is associated with low crystallographic symmetry and an insufficient number of slip systems; low chip strength; possible segregation of impurities; as well as poor workability at room temperature, which in turn prevents the widespread industrial use of titanium aluminides [8,9].

Currently, domestic and foreign experts take active actions to improve the properties of titanium aluminide-based alloys. Technical solutions are proposed that are aimed at improving the strength properties of materials, crack resistance, resistance to oxidation, and creep. These solutions are based on various approaches, including those related to alloying and heat treatment of alloys. Their practical implementation led to the development of alloys based on titanium aluminides of the second and third generations, which were successfully tested in industrial production [10,11].

Alloys based on titanium aluminide Ti_2_AlNb, which has an orthorhombic base-centered crystal lattice, belong to third-generation titanium aluminides [12,13,14]. This intermetallic compound has a wide homogeneity region, which allows forming titanium aluminides of various chemical compositions. Interest in these intermetallic alloys, which are called orthorhombic in the literature, is primarily due to improved mechanical properties and a good disposition of this phase to absorb hydrogen [15,16]. According to the results of research [15,16,17], the alloys of this system usually have a low density, and therefore they have a great advantage in achieving a potentially large hydrogen capacity. Of all titanium aluminides, intermetallic compounds based on the Ti_3_Al phase can contain and absorb a sufficiently large amount of hydrogen (4.3 wt.%). However, the use of such materials as hydrogen accumulators is hindered by their high temperature stability during desorption. Phases based on cubic and orthorhombic (B2, Ti_2_AlNb) lattices that appear when Nb is added to Ti_3_Al can solve this problem, since phases based on cubic and orthorhombic (B2, Ti_2_AlNb) lattices are “looser” compared to the close-packed structure based on fcc and fccp.

However, despite the production of a number of experimental alloys based on orthorhombic titanium aluminide with good mechanical and hydrogen storage properties, the practical application of these materials is still limited. First of all, this is due to the fact that, in the range of normal temperatures, titanium aluminides, like most intermetallic compounds, are characterized by increased brittleness in the polycrystalline state. One of the ways to solve this problem is the modification of alloy intermetallics based on Ti_2_AlNb with alloying elements such as vanadium, tungsten, molybdenum, and iron. It has been established that molybdenum additives increase the creep resistance of the material, tungsten-strength, and vanadium-ductility at room temperature [18,19,20,21]. However, the mechanical properties of these sintered parts are still unsatisfactory, and their production on an industrial scale has not yet been mastered.

This is due to the lack of reliable technological processes that would provide the necessary operational properties of the obtained materials [12,13]. There are also no experimental data on the behavior of these materials under aggressive conditions, for example, when exposed to high temperatures, the knowledge of which is extremely important for heat-resistant and refractory structural alloys. The complexity of the study of alloys based on titanium aluminide Ti_2_AlNb also lies in the fact that they can be multiphase, depending on the initial composition of the powder composition used. Thus, the analysis of current studies shows that there are still many questions regarding the fundamental thermodynamic and structural characteristics of intermetallic compounds (IMC). To date, this area is poorly understood and requires experimental research to study the influence of the parameters of sintering technological processes on the structure, phase state, and physical and mechanical properties of the IMC of the Ti-Al-Nb system.

Due to the foregoing, the aim of this research is to obtain an alloy based on Ti-Al-Nb systems with a bimodal structure and improved mechanical properties, for subsequent wide application, including as hydrogen storage materials.

This paper was devoted to studying the effect of high-temperature two-stage heat treatment on the structure and strength properties of the IMC of the Ti-25Al-25Nb at.% system.

## 2. Materials and Methods

The alloys studied in this paper are based on the Ti-24.5Al-24.5Nb at.% were obtained by combining the MA and SPS methods. Titanium powder with a particle size of 45–60 µm, niobium powder with a particle size of 40–63 µm, and aluminum powder with a particle size of 100–150 µm were used as starting materials. Table 1 provides the chemical composition of the powders used in the work.

Mixing and mechanical activation (MA) of the powder compositions (Table 2) was conducted using a planetary mill Retsch PM100SM; the ratio of the powder composition mass to the mass of grinding media was 1:10. The MA was conducted in grinding jars with a specially made lid providing an inert medium in the jar volume (argon, 4.5 × 10^3^ Torr). To prevent oxidation of the powder compositions after MA, the powders were removed from the grinding jars in a special box, where an inert medium was provided. Grinding media and a steel grinding jar with a protective jacket made of zirconium oxide were used to minimize contamination of powder mixtures during mixing and MA.

Consolidation of powder mixtures after MA was conducted in accordance with a previously developed method [22]. Spark plasma sintering (SPS) was performed using an SPS-515S unit at a temperature of 1300 °C, a static prepress pressure of 20 MPa, and a heating rate of 100 °C/min, and isothermal holding time was 5 min. Figure 1 shows a block diagram of the systematic combination of preliminary MA of a three-component powder composition and subsequent SPS.

Primary SPS experiments have been conducted in the temperature range of 1350–1500 °C, static prepress pressure—20 MPa, heating rate—100 °C/min, and isothermal treatment time was 5 min. In the obtained samples, melting of the aluminum component was observed, which has a negative impact on the quality of the products. Apparently, an increase in temperature from 1350 °C to 1500 °C during sintering of the powder mixture of the Ti-Al-Nb system leads to a sharp increase in the temperature of the Al particles in the mixture. Subsequently, due to the melting of the Al particles, it becomes impossible to control the phase formation process, which ultimately leads to the difficulty of obtaining the desired product. In this regard, further experiments on sintering were conducted at a temperature of 1300 °C.

In experiments on high-temperature treatment of samples with conditioning in an inert medium (argon), a high-temperature furnace LHT 02/16 with MoSi_2_ heating elements was used. In the experiment, a specially manufactured experimental ampoule device (EAD) was used, which is designed for heat treatment in an inert medium.

After heat treatment of EAD, samples were removed from the ampoule and subjected to cooling in water, after which they were placed in a plastic sealed package with marking.

Mechanical tests for three-point bending were conducted at room temperature on an Instron 5966 universal testing machine. The test was carried out until the test sample destructed at an active grip travel speed of 0.3 mm/min.

To ensure the possibility of installing and fixing the supports at the required distance between them, special devices were developed and manufactured (Figure 2). The distance between the supports for the samples was set equal to 16 h (14 mm). The calculation of strength characteristics was carried out in accordance with GOST R 56810-2015 “Flat Sample Bending Test Method”.

The microstructure and elemental composition of the resulting alloys were studied in the topographic contrast mode using a TescanVega3 scanning electron microscope (SEM) equipped with an X-ACT energy-dispersive spectral analysis (EDS) attachment.

Structural-phase states were studied using an Empyrean X-ray diffractometer. The operating mode of the PIXcel1D detector is a scanning line. Radiation: CuKα; voltage and current: 45 kV, 40 mA. A fixed divergence slit of 1° was used (distance from the divergence slit to the tube focus 87 mm), an anti-scattering slit of 2°, an incident beam mask marked 10, providing an incident beam width of 9.9 mm. Air temperature during shooting is 23 °C.

## 3. Results

### 3.1. SPS of Mechanically Activated Powder Compositions of the Ti-Al-Nb System

Figure 3 shows the curve of change in the linear dimensions of the Ti-Al-Nb samples during SPS at a temperature of 1300 °C, an analysis which allowed establishing the following features during consolidation of the test material. The shrinkage curves of the samples have a two-stage character. A high intensity of heat shrinkage under a pressure of 20 MPa is observed at relatively low temperatures (below the sensitivity limit of the pyrometer) and occurs in the temperature range of 535–585 °C (depending on the type of powder). Its intensity, with approaching the temperature of 575 °C, increases. In the temperature range of 635–840 °C (depending on the type of powder), shrinkage practically stops. At the same time, for powder mixtures (MA-20 and MA-180), the period of slow shrinkage is too short, which can be explained by melting and uniform distribution of the aluminum component in the volume. Then, up to a temperature of 1100–1200 °C (depending on the type of powder), the shrinkage intensity increases again and decreases significantly after a temperature of 1200 °C. It should be noted that the bulk of shrinkage occurs at the non-isothermal stage of heating. This indicates a high intensity of material consolidation.

According to the authors [23], during SPS, in the dynamics of linear shrinkage of reaction compositions based on Ti-Al-Nb at a temperature of 1200 °C, two endothermic peaks are observed in the temperature range of 850–1150 °C, which, according to the phase diagram of the Ti-22Al-25Nb at.% system [24], correspond to the phase transformation of IMC Ti_2_AlNb. S.L. Semiatin and others assumed that the first peak corresponds to the transition of the α2 + B2/β + O phases to the α2 + B2 phase, while the second peak corresponds to the transition of the α2 + B2 phase to the B2 phase. In our case, when consolidating elemental powders after the MA process, we observed a similar two-stage dynamic of linear shrinkage of sintered samples (Figure 3). However, the temperature boundaries of IMC phase transformations were smaller and took place in the range of 575–850 °C. This may be due to the preliminary MA, which led to an increase in the reactivity of the mixture. At the same time, it must be borne in mind that a high aluminum content can greatly shift the boundaries of phase transformations.

### 3.2. Structural and Phase State of Samples after Sintering

SPS at 1300 °C led to the formation of structures uniform in elemental distribution due to the almost complete dissolution and redistribution of niobium and titanium. The structure of the samples after sintering is predominantly bimodal, where the orthorhombic and cubic phases predominate, which are close in chemical composition to Ti_2_NbAl.

Observation of the microstructure of samples sintered at a temperature of 1300 °C revealed a number of features in the distribution and content of the main phases. This is evidenced by the images of the surface of the samples obtained using SEM (Figure 4a,b). As Figure shows, at a temperature of 1300 °C, almost complete diffusion of niobium (Table 3) and titanium occurs in the samples, while grain boundaries of the B2 phase are clearly identified on all samples.

According to the X-ray results (Figure 5a,b), for all samples obtained at the SPS temperature of 1300 °C, the main phases are the B2 phase and the O phase, and the presence of a small amount of the Ti_3_Al (α2) phase is also observed. Determining the volume fraction of the AlNb_2_ phase was difficult due to their low content and local distribution. The content of the O phase for the samples is the highest compared to other phases and reaches a maximum in the MA-180 sample, the value of which was 50.92%. Apparently, the O phase precipitates from the B2 and α2 phases, and an increase in the duration of the MA process has a positive effect on the IMC structure, namely, on the formation of the O phase.

The B2 phase with a bcc lattice has coincidences of the main lines with pronounced lines of other phases and is identified practically by one first line, which does not coincide with the lines of the other phases identified in the phase composition of the sample. In addition, this phase can be identified within a cubic primitive lattice with a parameter of about 0.330 nm. The line is clearly identified, reliably separated from the rest, and has a small width.

Thus, the main matrix phase for all samples after SPS is the B2 phase, with volumetric precipitation of the O phase at the boundaries and in the grain body of the B2 phase. The O phase has a chaotic distribution and morphology on the surface of the samples. As Figure 4 shows, the formation of lamellar and globular formations of the O phase is observed over the entire surface. At the same time, their sizes and shapes vary in a wide range of values from -nano to -micro. For example, on individual grains of the B2 phase, the presence of nanosized, and globular aggregations of the O phase were found (yellow arrow). The main difference in the surface structure of the samples is the formation of large lamellar precipitates, which is mainly inherent in the MA-20 sample.

At the same time, the presence of precipitates of α2 and AlNb_2_ phases was found in the microstructure of the samples. The separation of the α2 phase for the MA-20 sample is characterized by larger sizes up to 150 μm, while in the MA-180 sample their size does not exceed 50 μm. This indicates that during MA the powder mixture activated for 180 min has a higher reactivity. In this case, all samples are characterized by the formation of a barrier O phase, which envelops the precipitation of the α2 phase. Most likely, due to the higher diffusion activity in the MA-180 sample, at the sintering stage, large precipitates of the α2 phase decompose into many small globular inclusions. In addition, niobium/aluminum atoms occupy Ti positions in the B2 phase and form a barrier O phase around precipitates of the α2 phase.

It should be noted that the distribution of the AlNb_2_ phase on the sample surface at a sintering temperature of 1300 °C led to the formation of a coarse-grained dendritic structure from particles of this phase. The distribution of AlNb_2_-phase particles has two main features depending on the sample. The first feature is that accumulations of AlNb_2_-phase are observed on the surface of the MA-20 sample, in the form of short needle-like inclusions, without a specific orientation (Figure 4a). The second feature is that on the MA-180 sample, the main part of the AlNb_2_-phase particles has the form of plates, which in the field of the section look like needles up to 5 µm thick and 20–25 µm long. They are located mainly at the triple junctions of the B2-phase (Figure 4b) and grow like needles into the bodies of the grains and the boundaries of the B2 phases. At the same time, the area around the precipitates of the AlNb_2_ phase is characterized by the formation of the matrix B2 phase without precipitates of the O phase. At the same time, an increased content of Nb is observed in this region. A more detailed analysis of the evolution of the microstructure of mechanically activated powder mixtures and the mechanism of formation of secondary phases depending on the temperature of SPS was described by the authors in [25,26,27].

It should be noted that the study of the fine structure of the samples made it possible to determine the formation of a special type of mutual arrangement of precipitates of a coarse-grained dendritic structure from particles of the AlNb_2_-phase; Figure 6 shows a section of the fine structure of the alloy, where an acicular plate of the AlNb_2_-phase is seen with AlNb_3_-phase precipitates in its body (indicated with dotted lines and arrows) formed during sintering.

In [28], the authors presented calculations showing that the formation of ensembles of regularly arranged crystals of a new phase inside the original phase is one of the ways to reduce the stress fields in it. For this kind of ensembles of needle-like plates, the term “truss-like” groups (trusses) is used. In the case of such a farm, if the energy of the total field of the connection of two crystals is less than the total energy of the fields of each crystal, such a connection is energetically favorable. Forming a truss, these plates change the state of the B2-matrix, as a result of which the growth of neighboring combined AlNb_2_-phase plates turns out to be interconnected, and similar trusses are formed.

### 3.3. Structural and Phase State of Samples after Heat Treatment

In the microstructure of the IMC samples of the Ti-Al-Nb system, obtained by combining MA and SPS, there was a large scatter in the sizes and shapes of particles of the identified O phase. In addition, residues of unreacted niobium and a bulk number of secondary phases such as Nb_2_Al and α2 were found, which is mainly due to the short SPS holding time and subsequent slow cooling. In order to avoid the negative consequences of the presence of secondary phases (decrease in the strength of the alloys), as well as in order to obtain a predominantly bimodal O + B2 microstructure, work was carried out on heat treatment. The selection of heat treatment modes was carried out primarily on the basis of state diagrams presented by the authors of [29], for alloys based on the Ti-Al-Nb system. First of all, it should be noted that the state diagram used was tentative, since the nonequilibrium system of a mechanically activated powder mixture could greatly shift the temperature boundaries of phase transformations.

According to the state diagram [29], heat treatment was conducted in two stages—at 1250 and 800 °C for 2 h (Figure 7). The treatment time was chosen to be 2 h, since, according to the authors in [30], heat treatment at a temperature of 1250 °C with a treatment of 2 h ensures complete diffusion of unreacted niobium due to an increase in its diffusion rate by approximately four times. During heat treatment at temperature T_1_, a single-phase B2 structure is formed. Cooling from temperature T_1_ to T_2_ leads to the decomposition of the B2 phase into O and α2 phases. The subsequent two−hour treatment at a temperature of T_2_ contributes to the stabilization of the resulting bimodal (O + B2) -structure.

At a given temperature, part of the B2 phase decomposes into the α2 and O phase during conditioning [31], where the α2 phase subsequently transforms into the O phase [32]. Final abrupt cooling in water regulates the thickness of the plates of the primary O phase.

It is noted in [33,34] that dissolution and diffusion of Nb during the α2→O transformation are of primary importance. Thus, the Nb-saturated regions are converted into the O phase, while the lack of Nb leads to the precipitation of the α2−phase. The authors of [35] believe that these transformations are associated with supersaturation of the α2 phase with niobium, which subsequently transform from the α2−phase with a hexagonal lattice into the O−phase.

In [36], the authors present the results of an energetically favorable transformation of the Ti_3_Al-Nb system, where with an increase in the content of niobium in the Ti-25Al-12.5 Nb at.% alloy, the free energy of the system increases, which is accompanied by a change in the type of lattice—from hcp to orthorhombic one. This trend, apparently, can also be predicted for alloys with a niobium content above 12.5 at.%.

On the diffraction patterns of the X-ray samples after annealing, the lines of the O and B2 phases were clearly defined. The phase composition of samples MA-180 and MA-20 after heat treatment is characterized mainly by a bimodal structure. After heat treatment, the presence of the line of Nb_2_Al-phase peaks are not observed on the diffraction patterns. The results of X-ray of samples after two-stage heat treatment are shown in Figure 8 and Table 4.

Thus, the main phases in all compositions are the orthorhombic phase of the O-AlNbTi_2_ type and the cubic B2 phase. The lattice parameter of the cubic phase is a = 0.323 nm in all samples. After annealing, there are no identifiable phase lines of the AlNb_2_ or Nb in the phase composition of the samples. At the same time, the X-ray diffraction pattern of a sample of the MA-20 composition contained deviations in the peak profiles of the main phases identified, which could be assigned to an α2-based or α-Ti based solid solution. An important feature of the diffraction patterns from the surface of the blanks was the good correspondence between the relative intensities of the peaks of the detected phases and the reference ones, which indicated the absence of texture and large crystallites.

Figure 9 shows the microstructures of the samples after two-stage heat treatment. As shown in the figure, after processing, the microstructure of the samples is characterized by a predominantly bimodal structure. In addition, high-temperature treatment, according to X-ray data, led to an increase in the content of O-phase precipitates.

The results of X-ray and microstructural analysis confirm the absence of unreacted niobium and the Nb_2_Al phase in the bulk of the samples. However, in some areas, the presence of globular and line segregations of the α2 phase was found, which are mainly located at the grain boundaries of the B2 phase (Figure 9). In addition, the distribution of the O phase has become more uniform, both in size and shape. A large number of globular and lamellar particles of the O phase precipitated inside and at the grain boundaries of the B2 phase, outlining them along the perimeter. The average thickness of lamellar precipitates of the O phase is 250–500 nm, while their length is 10 times greater than the width. The size of individual large precipitates of the lamellar O phase in local areas reaches 5 μm. The average grain size of the B2 matrix after heat treatment increased by 20% and is about 80 µm, as shown in Figure 9. Thus, after treatment, only the B2 and O phases are observed, where the O phase becomes dominant in the microstructure of the alloy sample. The precipitates of the O phase are evenly distributed in the body of grains of the B2 phase and are randomly located relative to each other.

The images of the fine structure of the samples presented in Figure 10 indicate that the structure of the investigated alloy after heat treatment has a lamellar morphology, with local precipitates of the α2 phase.

Moreover, according to the results of TEM studies, plates with a similar contrast were found. However, the interpretation of microelectron diffraction patterns showed that areas with a similar contrast have a different lattice type; in Figure 11a–c, the elongated dark-colored plates are O phase particles. The third type of plates found in the structure, which has a similar contrast, is a package of parallel plates of the α2-phase (Figure 11d). Obviously, the described types of structures are associated with different stages of the α2 → O-transformation in the studied alloy.

During heat treatment, due to the redistribution of niobium in the plates of the α2 phase, the content of niobium increases and at a certain moment its concentration reaches a critical point, as a result of which the lattice type changes from an ordered hcp to an orthorhombic one with the formation of an O phase. The growth of the O-phase nucleus is accompanied by depletion of the α2-phase plates in niobium and a decrease in size while maintaining the hcp lattice. Thus, due to the redistribution of niobium, large precipitates of the α2 phase are broken up into globular and line precipitates of small sizes of the α2 phase, which are deposited mainly at the grain boundaries of the matrix B2 phase.

### 3.4. Strength Properties of Samples before and after Heat Treatment

Figure 12 shows the SEM images of samples surface morphologies. In samples sintered at 1300 °C, cleavage is the main type of fracture, which is indicative of the poor plasticity of the alloy. The sample surface after SPS has a mixed fracture morphology with a quasi-cleavage and depressions. In addition, the fracture surface of the samples has a flat fracture surface; such a fracture surface morphology indicates that there was almost no slippage of dislocations. Both samples, regardless of the MA duration, are characterized by low strength and ductility. This is due to the volumetric amount of large precipitates and uneven distribution of brittle α2 phases.

In addition, at the given sintering temperature, many acicular precipitates of the O phase began to grow from the boundaries to the body of grains of the B2 phase. Crystalline defects of grain boundary phases can reduce the critical energy of new phase nucleation. Therefore, the new O phase (secondary acicular O phase) prefers nucleation at grain boundaries rather than inside B2 grains. The nucleation and growth of this acicular O phase caused a strong strain concentration, which further led to the failure of bending tests and low strength values for the samples.

After heat treatment, both samples show a higher yield strength, reaching 1310 MPa (MA-180) and 1322 MPa (MA-20). The surface fracture of the samples obtained after testing is characterized by intergranular (intercrystalline) brittle fracture with “river” or “step” features. This trend is associated with the crystal bimodal structure of the samples, where the B2 phase with a bcc lattice is the main matrix structure. The crystal structures of the two alloys are typical bcc structures. In the bcc crystal structure, the critical stress of the sliding drive system is larger, and the number of sliding systems is smaller, and therefore the samples after testing show the performance characteristics of high strength and low ductility.

As Figure 13 shows, with a similar fracture pattern, the crack passes along the grain boundaries of the B2 phase (yellow arrows). At the fracture, individual grains are observed in the form of intergranular facets having the shape of polyhedrons. In addition, the fracture surface of individual B2 phase grains has a mixed morphology with blurred areas, as evidenced by many tiny depressions indicating a ductile fracture.

## 4. Conclusions

In the present work, the effect of two-stage heat treatment on the change in the structural-phase state and mechanical properties of alloys of the Ti-Al-Nb system with a bimodal structure was studied. Based on the results obtained, the following conclusions can be drawn:SPS of mechanically activated powder mixtures of the Ti-Al-Nb system at a temperature of 1300 °C and a static prepress pressure of 20 MPa with an isothermal holding for 5 min allows obtaining an alloy with a predominantly bimodal (O + B2) structure;After SPS, the main structural-phase composition is the matrix B2-phase with a cubic lattice and the volumetric amount of precipitates of the orthorhombic O phase at the boundaries and in the body of grains of the B2-phase. At the same time, traces of a volumetric number of secondary phases are still observed in the structure, such as: AlNb_2_, α2, and unreacted Nb;Mechanical properties of alloys of the Ti-Al-Nb system obtained by combining the MA and SPS processes are characterized by abnormally low values of strength (less than 150 MPa) and ductility. This is due to the formation of a large number of large precipitates of the brittle α2-phase, as well as a strong concentration of deformation at the boundaries of the matrix B2-phase, caused by the nucleation and growth of the acicular O phase;Two-stage heat treatment of alloys of the Ti-Al-Nb system leads to a decrease in the size of the precipitates of the O phase and to an increase in its content while maintaining the bimodal structure. After heat treatment, the O phase becomes dominant in the structure of the alloys, while the AlNb_2_ and Nb phases are completely dissolved. Coarse precipitates of the α2-phase are transformed into nuclei of the O phase due to saturation with niobium, fine precipitates of the α2-phase depleted in niobium precipitate in the form of globular and line precipitates at the boundaries of the B2-phase;The strength and ductility of Ti-Al-Nb alloy samples after heat treatment increases by 10 (1310 MPa, MA-180) and 2 (1322 Mpa, MA-20) times, respectively. The surface fracture of the samples obtained after testing is characterized by intergranular (intercrystalline) brittle fracture with “river” or “step” features. In a crystal structure with a bcc lattice, the critical stress of the sliding drive system is greater, the number of sliding systems is less, and therefore, the samples after testing demonstrate the performance characteristics of high strength and relatively low ductility.

## 5. Patents

Y.A. Kozhakhmetov, E.G. Batyrbekov, M.K. Skakov, et al. Innovative Patent of RK No. 5809 “Method for producing hydrogen-storage rechargeable intermetallic compounds”, bul. No. 4, print. 29 January 2021.

## Figures and Tables

**Figure 1 materials-15-05560-f001:**
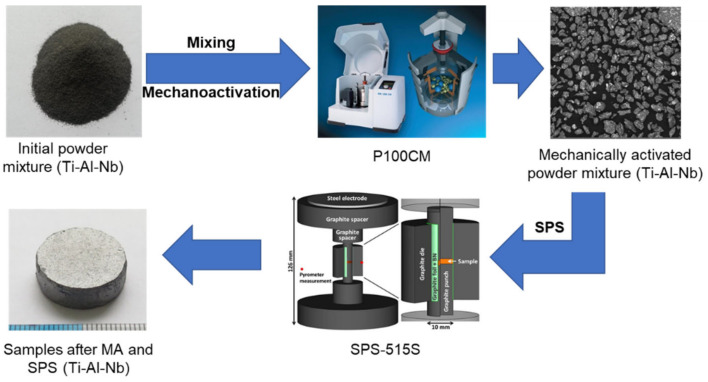
Scheme of combining technological processes of SPS to obtain IMC of the Ti-Al-Nb system.

**Figure 2 materials-15-05560-f002:**
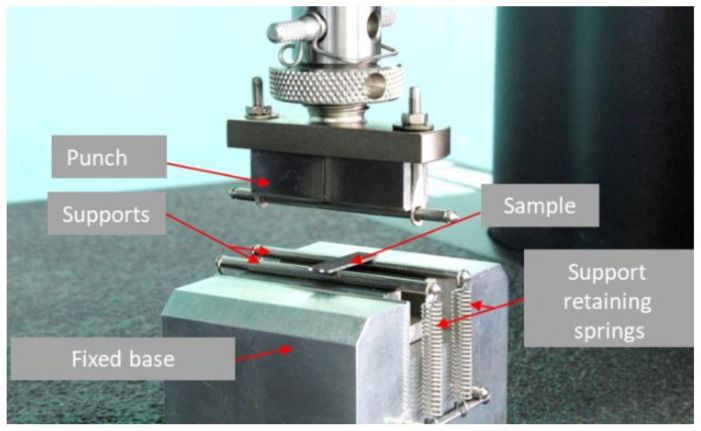
Outer view of the device for three-point bending.

**Figure 3 materials-15-05560-f003:**
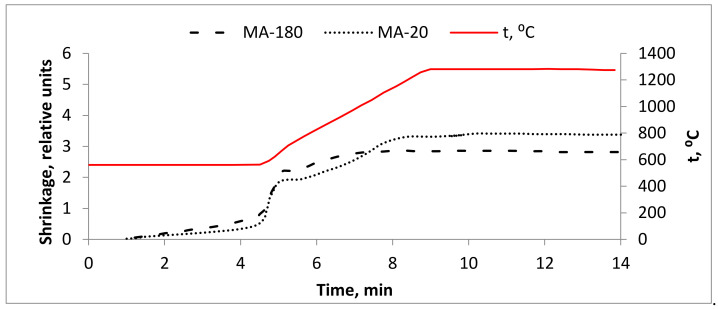
Dynamics of linear shrinkage of reaction compositions based on Ti-Al-Nb in the process of SPS at a temperature of 1300 °C.

**Figure 4 materials-15-05560-f004:**
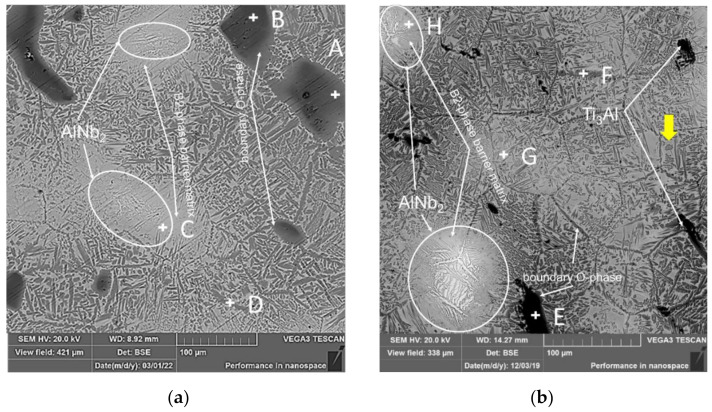
SEM image of samples based on Ti-Al-Nb system after SPS under the conditions of 1300 °C, 5 min, 20 MPa: (**a**) MA-20; (**b**) MA-180.

**Figure 5 materials-15-05560-f005:**
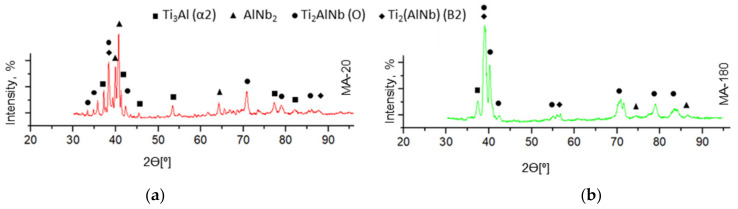
The X-ray results of samples sintered at 1300 °C/5 min/20 MPa: (**a**) MA-20; (**b**) MA-180.

**Figure 6 materials-15-05560-f006:**
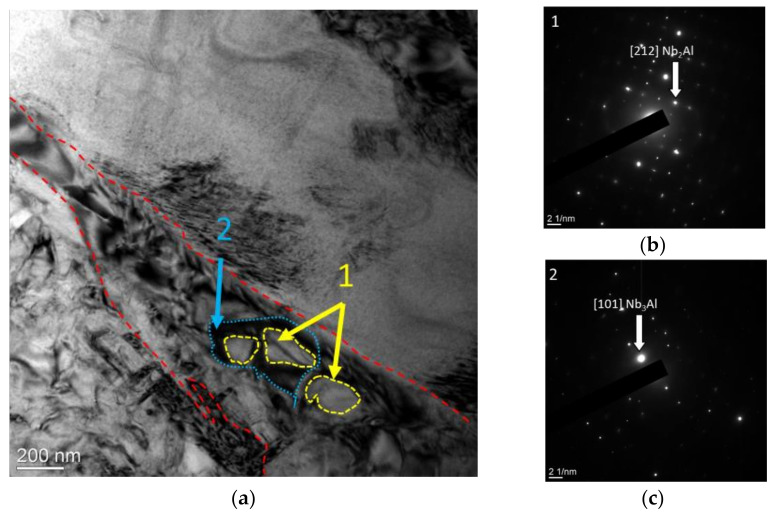
Fine structure of the Ti-25Al25Nb at.% system alloy: (**a**) bright field; (**b**,**c**) electron diffraction pattern from selected areas.

**Figure 7 materials-15-05560-f007:**
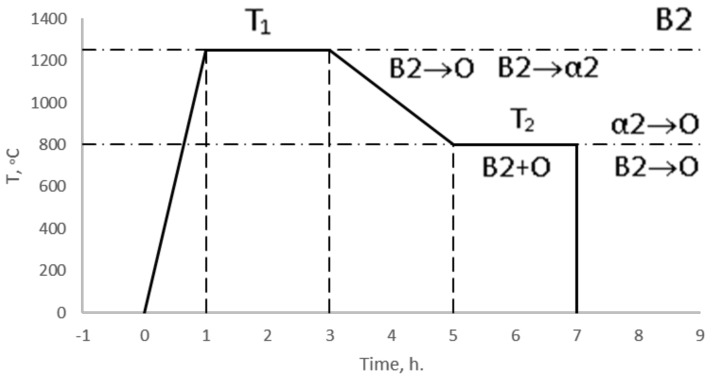
Temperature—time scheme for processing samples of the Ti-Al-Nb system.

**Figure 8 materials-15-05560-f008:**
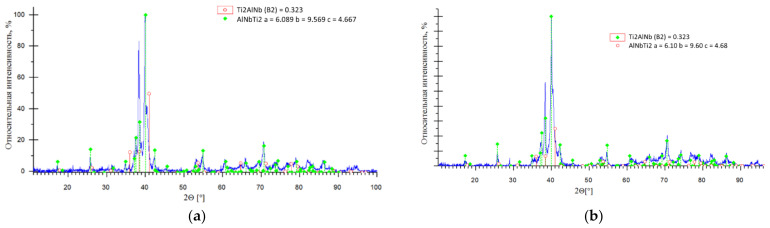
X-ray results of annealed samples: (**a**) MA-180; (**b**) MA-20.

**Figure 9 materials-15-05560-f009:**
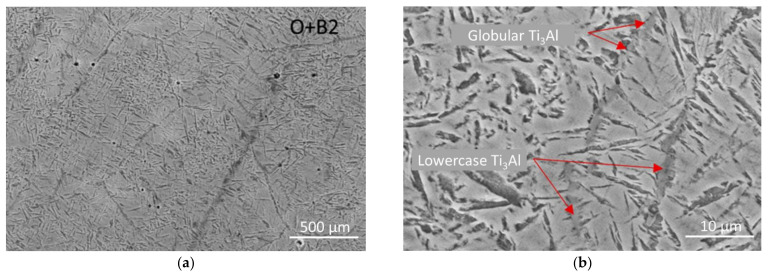
Structure of IMC samples of the Ti-Al-Nb system after two-stage heat treatment: (**a**) ×500; (**b**) ×2500.

**Figure 10 materials-15-05560-f010:**
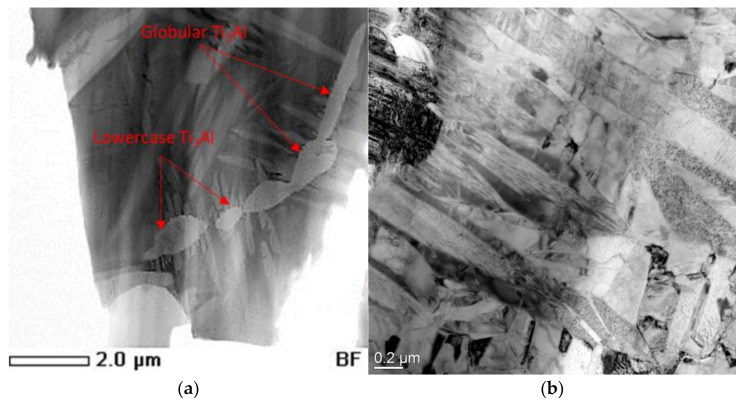
Fine structure of the alloy of the Ti-25Al25Nb at.% system: (**a**) STEM; (**b**) TEM.

**Figure 11 materials-15-05560-f011:**
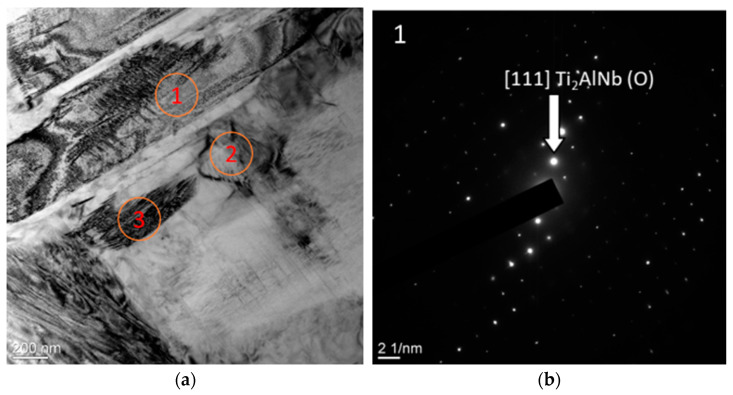
Fine structure of the alloy of the Ti-25Al25Nb at.% system: (**a**) bright field; (**b**–**d**) electron diffraction pattern from selected areas.

**Figure 12 materials-15-05560-f012:**
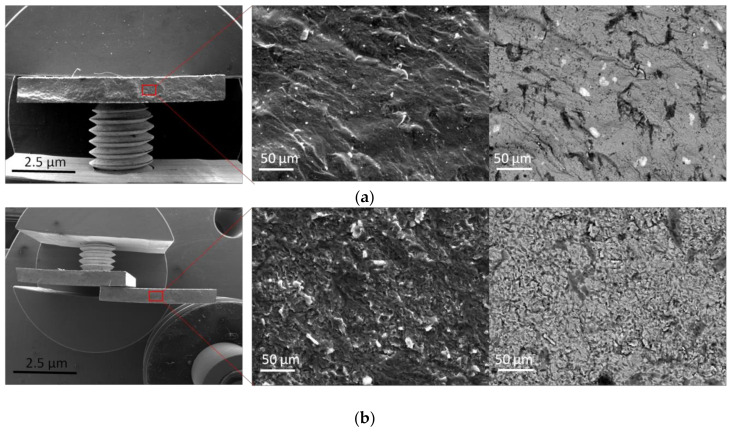
Factography of titanium aluminide samples: (**a**) Morphology of the fracture surface of sample MA-20; (**b**) Morphology of the fracture surface of sample MA-180.

**Figure 13 materials-15-05560-f013:**
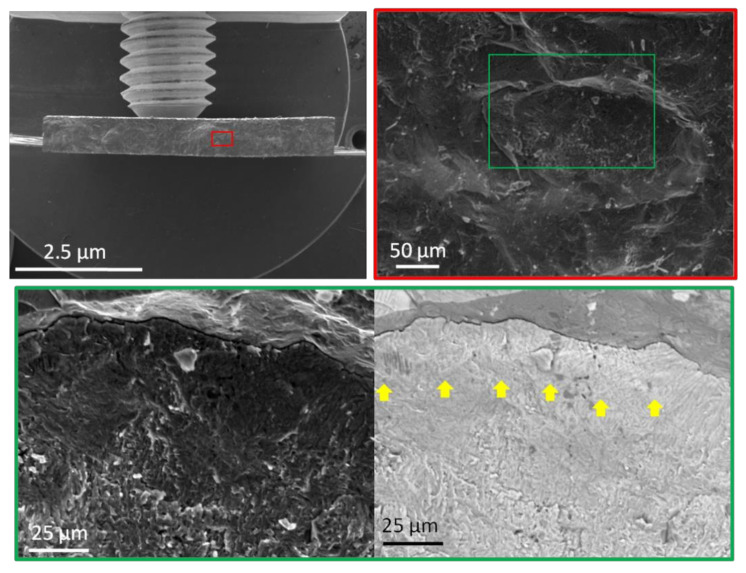
Fractography of titanium aluminide samples after heat treatment.

**Table 1 materials-15-05560-t001:** Chemical composition of the powders it.

Chemical Composition, Impurities, wt.%, Maximum									
Powder	N	C	H	Fe + Ni	Si	Cu	Ta	W	O
Ti	0.08	0.05	0.35	0.04	0.01	0.004	-	-	-
Nb	0.02	0.005	0.01	0.004	0.003	-	0.06	0.003	0.08
Al	-	-	≥0.2	0.35	0.4	0.02	-	-	-

**Table 2 materials-15-05560-t002:** MA parameters.

Material	Reference Designation	Duration, Min.	Rotating Speed, rpm	Test Medium
Ti-Al-Nb mixture	MA-20	20	650	Argon
	MA-180	180	350	

**Table 3 materials-15-05560-t003:** Results of local elemental analysis of the surface (at.%).

Name	Al	Ti	Nb	Phase	Name	Al	Ti	Nb	Phase
A	14.17	65.7	20.13	O	E	27.12	67.7	5.18	α2
B	21.11	69.96	8.93	α2	F	24.73	53.17	22.1	O
C	14.27	56.97	28.76	B2	G	28.4	58.57	1.04	B2
D	25.83	51.03	23.14	O	H	14.81	63.93	21.26	B2

**Table 4 materials-15-05560-t004:** Results of X-ray diffraction phase analysis.

Sample	Main Phases, Lattice Parameters, nm	Other Phases, Lattice Parameters, nm
MA-2S	o-AlNbTi_2_(CmCm; a = 0.689 b = 0.956 Ti_2_AlNb (B2) (Pm-3m; a = 0.323);	(Ti,Nb,Al)C (Im3m, a = 0.427); Nb(Im3m; a = 0.330); α2 (a= 0,586, c = 0.470);
MA-1P	o-AlNbTi_2_(CmCm; a = 0.610 b = 0.960 c = 0.468); Ti_2_AlNb (B2) (Pm-3m; a = 0.323);	

## Data Availability

The data used to support the findings of this study are available from the corresponding author upon request.

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
