# Peer review of "Effect of a High-Temperature Treatment on Structural-Phase State and Mechanical Properties of IMC of the Ti-25Al-25Nb at.% System"

_materials, 2022, doi:10.3390/ma15165560_

Round 1

Reviewer 1 Report

The present manuscript explores the effect of two-stage heat treatment on the change in the structural-phase state and mechanical properties of alloys of the Ti-Al-Nb system with a bi-modal structure. It is found that the mechanical properties of the Ti-Al-Nb system obtained by combining the 404 MA and SPS processes exhibits abnormally low values of strength (less than 405 150 MPa) and ductility due to the formation of a large number of large precipitates 406 of the brittle α2-phase, as well as a strong concentration of deformation at the boundaries 407 of the matrix B2-phase, caused by the nucleation and growth of the acicular O-phase. Further two-stage heat treatment of alloys of the Ti-Al-Nb system leads to a decrease in the 409 size of the precipitates of the O-phase and to an increase in its content while maintaining 410 the bimodal structure  which imporves the tensile strength and ductility of Ti-Al-Nb alloy significantly. This is a well-organized manuscript which is worth for publication.

1) Refs. 12 and 19 are repeated, one of them has to be replaced by Vacuum  161, 2019, 209.

2) There are numerous publications on the microalloying of Ti2AlNb-based alloys. These have to be shortly summerized in the introduction, for example,  Mo-modified Ti2AlNb alloys in Materials Science and Engineering: A   776, 2020, 139043, Journal of Alloys and Compounds  842, 2020, 155794, 876, 2021 2021, 160110... This is important for the author getting a full overview for the recent progress in this field.

3) Figure 12c and Fig.13b are not necessary for the reason that no additional information except for the ultimate strength can be recognized from the measured curves.

Reviewer 2 Report

The manuscript entitled: Effect of a High-Temperature Treatment on Structural-Phase State and Mechanical Properties of IMC of the Ti-25Al-25Nb system at. % deals with the fabrication of Ti-Al-Nb alloy with a bimodal structure with improved mechanical properties for various applications. I have the following concerns with the present manuscript.

- The aim of the manuscript is not clear.

- Why Ti-Al-Nb system is connected with hydrogen storage application? It should be explained in detail.

- Table 1 - Chemical composition of the powder - is it wt.% or at.% or vol.% ?

- How do you ascertain the powder after MA is free from impurities? Any proof?

- All the experimental details should be in section 2. Materials and Methods and not in section 3. Results.

- Scale bars in Fig. 12, 13 are not visible.

- A strong scientific discussion is missing.

- Typos in the manuscript need to be rectified carefully. For instance, in 2. Materials and Methods section - 2nd para - 4th line 103 torr should be 10(pow)3 torr.

- The English language needs attention.

Round 2

Reviewer 2 Report

The authors have managed to answer the issues raised. Even though they are not satisfactorily addressed, I may reluctantly recommend the manuscript for publication.